# Adoption Can Be a Risky Business: Risk Factors Predictive of Dogs Adopted from RSPCA Queensland Being Returned

**DOI:** 10.3390/ani12192568

**Published:** 2022-09-26

**Authors:** Eileen Thumpkin, Mandy B. A. Paterson, John M. Morton, Nancy A. Pachana

**Affiliations:** 1School of Psychology, The University of Queensland, Brisbane, QLD 4072, Australia; 2Royal Society for the Prevention of Cruelty to Animals Queensland, Brisbane, QLD 4076, Australia; 3School of Veterinary Science, The University of Queensland, Gatton, QLD 4343, Australia; 4Jemora Pty Ltd., East Geelong, VIC 3219, Australia

**Keywords:** dog adoption, RSPCA Queensland, shelters, pet relinquishment, returned adoption, risk factors, companion-animal relinquishment, companion-animal surrender, characteristics, surrender reasons

## Abstract

**Simple Summary:**

Current knowledge about the dogs in care and their adoption needs provides shelters with the information to promote and match dogs more successfully with potential adopters. The analyses reported here provide up-to-date insights about the dogs adopted from the Royal Society for the Prevention of Cruelty to Animals Queensland (RSPCA Queensland) and risk factors that influenced their risk of readmission. For the dogs adopted, age and body weight at adoption, days in foster before adoption, and colour and breed are all independently associated with the risk of readmission to RSPCA Queensland.

**Abstract:**

Not all dog adoptions are successful. This two-year retrospective study used survival (i.e., time-to-event) analyses to investigate readmissions for dogs adopted from RSPCA Queensland shelters between 1 January 2019 and 31 December 2020. A better understanding of temporal patterns and risk factors associated with readmission may help RSPCA Queensland shelters better target and tailor resources to improve retention by adopters. The failure function (the cumulative percentage of adoptions that were readmitted by day of the adoption period) increased rapidly during the first 14 days of the adoption period. Approximately two-thirds of all returns occurred in this period. This readmission rate may have been influenced by the RSPCA Queensland adoption-fee refund policy. The cumulative percentage of adoptions that were readmitted plateaued at just under 15%. Dog size, age, coat colour, breed, and spending time in foster before adoption were factors associated with the risk of readmission. Failure functions for a low and a high-risk adoption example demonstrated the large degree of difference in hazard of readmission between covariate patterns, with estimated percentages of adoptions being returned by 90 days for those examples being 2% and 17%, respectively. Spending time in foster care before adoption appears to be protective against readmission, presumably because it supports a successful transition to the new home environment. Behaviour support and training provided for dogs during foster care may contribute to improve their outcomes. These findings highlight the profile of the higher-risk dogs potentially providing shelters with an opportunity to examine where and how resources could be allocated to maximize outcomes for the overall cohort. Population attributable 90-day failure estimates were calculated for each of bodyweight and age at adoption, coat colour, spending time in foster care before adoption, and breed. This calculation shows the expected reduction in the cumulative percentage of dogs readmitted by day 90 if the hazards of readmission for higher risk categories were reduced to those of a lower risk category. Expected reductions for individual factors ranged from 1.8% to 3.6% with one additional estimate of 6.8%. Risk of readmission could be reduced through increased development of foster capacity and capability, targeted interventions, improved adopter-dog matching processes, and more effective targeting of support for higher risk dogs, such as older or larger dogs. Population impact analyses provide a macro view that could assist shelters in strategically assessing the return on investment for various strategies aiming to improve adoption outcomes and potentially reduce readmissions.

## 1. Introduction

Welfare and rescue organisations worldwide play a critical role in caring for and rehoming surrendered and stray companion animals. Published estimates suggest that millions of animals come into the care of these organisations annually. For example, the American Society for the Prevention of Cruelty to Animals (ASPCA) reported that an estimated 6.5 million animals entered shelters nationwide in 2019, of which 3.1 million were dogs [1]. The DogsTrust [2] in the United Kingdom (UK) took in 14,301 dogs in 2019, while Cosgrove, in 2022 estimated that 2.7 million animals enter UK shelters each year of which 664,000 are dogs [3]. The Royal Society for the Prevention of Cruelty to Animals (RSPCA) Australia [4] reported in 2021 that its shelters received 103,057 animals, of which 22,311 were dogs and Fraser [5] reported recently, that approximately 200,000 dogs enter shelters and municipal rescue shelters annually. Significant numbers of these animals are reunited or rehomed, and encouragingly euthanasia rates of adoptable animals have been declining over the past decade [1,4,6,7,8,9].

Not all adoptions from rescue organisations result in successful, long-term homes for the dogs. Reported surrender rates for unsuccessful adoptions range between 7–20% [10,11,12,13]. Variability in estimates arises from the different definitions used by shelters, for example, return, surrender, or relinquishment, inconsistency in data entry and differing shelter intake policies. Several studies report that a significant percentage of the readmitted dogs came back within the first two weeks, some lasting only 24 h in their new home [8,14,15,16]. It is also important to note that owners may not return their dog to a shelter. Dogs could be given to family and friends, sold or euthanized.

An unsuccessful adoption of a companion animal impacts both the people and animals. Many owners report that giving up their dog was a very difficult and traumatic decision [16,17]. Research shows that time in a shelter can also be stressful for the dogs and may lead to deterioration in their health and behaviour [18,19,20]. Subsequently, the management and care of the animal once in care puts immense pressure on an organisation’s often limited resources and caring for these animals can negatively impact carers, volunteers, shelter staff, and ultimately the outcomes for the surrendered animals [7,13,21].

Preventing owner surrender of companion animals is a persistent challenge for animal shelters [22,23,24]. Within the growing body of adoption and relinquishment literature, results from numerous studies reveal a complex interplay of factors that reportedly influences successful and unsuccessful adoption outcomes [15,25,26,27,28]. Relinquishment can be unavoidable and should not be seen as an adoption failure. Readmissions provide shelters with an opportunity to learn more about the dog, which hopefully improves the dog’s chances of long-term retention. The ideal outcome, however, is to keep dogs in their adoptive home where appropriate, and to support adopters to resolve solvable factors and prevent relinquishment. Improving first time adoptions and thus decreasing the rate of readmissions delivers benefits on many levels. The individual dog and adopter benefit, those who care for the dog succeed in their endeavours to provide a long-term home, shelter capacity and resources can be reassigned to assist more animals in need, and the broader community and animal management authorities also incur fewer costs.

Despite the significant body of literature investigating and documenting factors associated with relinquishment of companion animals, and adoption of relinquished dogs, there are fewer studies that focus on the outcomes for dogs after adoption, specifically their possible return to the shelter post adoption, and the risk factors influencing this outcome.

This study aimed to describe temporal patterns of readmission of dogs after adoption from RSPCA Queensland shelters, to identify risk factors for readmission of adopted dogs, and to estimate the maximum population impacts of strategies that target high risk factors and aim to reduce the risk of readmission.

## 2. Materials and Methods

### 2.1. Study Overview

This study was a retrospective single cohort study using a cohort of dog adoptions from RSPCA Queensland over a two-year period. All dog adoptions from the eleven RSPCA Queensland rescue centres and six offsite events or retail outlets on and between 1 January 2019 and 31 December 2020 were enrolled, and readmissions to RSPCA Queensland up to 1 April 2021 identified.

### 2.2. Data Collection

Admission and readmission data for the study period were sourced from the RSPCA Queensland database, ShelterBuddy^®^. Data collected for each adoption included the dog’s unique identification number (ID), details for the admission preceding the adoption (admission date, source, reason for relinquishment if source was owner relinquishment), site of adoption, sex, coat colour, predominant breed, age and bodyweight at adoption, behavioural and health details, time in foster, time in shelter, adopter ID and postcode, and owner reasons for return after adoption (if returned). Neuter status was not included in the study as RSPCA policy requires all dogs are desexed prior to adoption.

Time-to-event methods used in this study involve the analysis of times from adoption to readmission and offer the advantage of including the dogs, who had not been readmitted during the study follow-up period in analyses. This inclusion avoids major selection bias and interpretation difficulties if only readmitted dogs had been analysed. It also allows assessment of readmissions patterns for more extended periods after adoption than if readmission status at a specified time (e.g., readmitted or not readmitted by 90 days after adoption) is assessed.

ShelterBuddy^®^ admissions data were searched for readmissions up to and including 1 April 2021. This ensured that each adoption was monitored (followed up) for subsequent readmission to RSPCA Queensland until at least day 92 of their adoption (where the day of adoption is day 1).

Approval to access the data on the dogs was granted by the Chief Executive Officer and the Board of RSPCA Queensland.

### 2.3. Preparation of Exposure Variables

Data from ShelterBuddy^®^ were imported into Microsoft Excel. Initially, these data sets, some of which included large numbers of sub-categories for example, coat colour, and breed label, were reviewed by the team, individually and then as a group. Selection was informed by a pilot study and previous work within the organization as well as the characteristics reported in published studies on adoption and relinquishment. From this initial review of the data, twenty putative risk factors were identified and agreed to by consensus for analyses. Exposure variables were then generated for each. Shelter staff were consulted to clarify definitions and to refine categories for exposure variables with numerous categories. This refinement was particularly complex for breed and coat colour, as these variables had 124 and 73 unique descriptions, respectively. Breed identification based on visual assessment can be difficult and complex, especially for mixed breed dogs [27,29,30]. It is acknowledged this may lead to a level of misclassification. For most adoptions in the current study population, the dog was recorded as mixed breed, with less than 3% of adoptions recorded as being purebred dogs. After reviewing the dog breeds cited on the American Kennel Club, the Australian National Kennel Club Council Ltd. and Pets4Life websites, the 124 breed descriptions were collapsed into 11 categories and labelled based on the morphologically predominant breed features. Breed categories with few adoptions were pooled into ‘other’ for analyses. The 73 coat colour descriptions were collapsed into 12 categories, consisting of six solid colours: black, white, fawn, tan, red and chocolate, five patterned coat categories: merle, tricolour, brindle, sable, and roan, and a single pooled category for coat colours with low numbers of adoptions.

Other exposure variables selected for analyses were: the dog adoption number (adoption numbers ranged from 1–5 where 1 indicated the first adoption for the dog during 2019 or 2020, 2 indicated the dog’s second adoption during that period and so on), sex of dog, source of admission immediately prior to the study adoption, reason for relinquishment if source was owner relinquishment, prior number of admissions to RSPCA, prior exits from RSPCA not as adoptions (mostly transfers to other rehoming organisations), prior number of adoptions from RSPCA Queensland (the 3 latter variables assessed on or since 1 January 2011), calendar month of adoption, adoption from an RSPCA offsite event or from an RSPCA shelter, days in foster care between admission and adoption, days available for adoption since admission, adoption year/in COVID lockdown period, and whether the dog was under behaviour management for at least some of its time with RSPCA from admission to adoption, whether a behaviour consultation with the adopter was required at time of adoption and/or during its time with RSPCA between admission and adoption, and whether the dog had been temporarily classified as disposition under final review (i.e., was reviewed for euthanasia due to serious behavioural attributes) during its time with RSPCA.

The Index of social disadvantage for the adopter’s postcode area was also assessed as a putative risk factor. Index of social disadvantage scores for 2016 (released 27 March 2018) were obtained from the Australian Bureau of Statistics (ABS; https://www.abs.gov.au/statistics last accessed on 19 September 2022). This index ranks geographic areas (defined by postcode) in Australia according to relative socio-economic disadvantage; a lower score indicates that an area is relatively disadvantaged compared to an area with a higher score. Decile numbers were used where the lowest 10% of areas nationally based on score were in decile 1, and the highest 10% of areas were in decile 10.

### 2.4. Readmissions

Times from adoption to readmission to RSPCA Queensland were calculated, where dogs readmitted on the same date as the date that they were adopted were allocated a time of 1 day, dogs readmitted on the following day were allocated a time of 2 days, and so on. For adoptions where the dog was not readmitted to RSPCA Queensland by the end of the last day of the study follow-up period (1 April 2021), their time from adoption to readmission was right censored on that date.

### 2.5. Statistical Analyses

Statistical analyses were performed using Stata (version 16, StataCorp, College Station, TX, USA). Time-to-event analyses were used.

The concept of hazard of readmission is central to these time-to-event analyses. As time was divided into discrete intervals (days), the hazard of readmission for a particular day described the probability of a dog being readmitted during that day given that it had not been readmitted before that day.

The failure function (the probability of being readmitted by time from adoption) was calculated using the Kaplan–Meier product-limit method with Stata’s -sts graph- command. Point-wise confidence intervals were calculated using the asymptotic variance as described by Kalbfleisch and Prentice [31]. The smoothed hazard function was also generated using Stata’s -sts graph- command, as the weighted kernel-density estimate. Pointwise confidence bands for smoothed hazard functions were calculated using the method based on a log transformation as described by Klein and Moeschberger [32]. These confidence intervals are only approximate as they did not account for lack of independence of repeated adoptions of the same dog.

Associations between the putative risk factors and time to readmission were assessed using Cox proportional hazards models, fitted using Stata’s -stcox- command. For categorical exposure variables, these models compare hazards between subgroups of subjects and estimate the ratio of these hazards as hazard ratios (HR). Overall *p*-values for variables were calculated using likelihood ratio tests. All adoptions in 2019 and 2020 were used to assess the association between the dog’s adoption number and time to readmission. Standard errors of coefficients for adoption number were adjusted to account for clustering of adoption within dog for this analysis by use of cluster-robust standard errors. For all other analyses, only each dog’s first adoption in the period from 1 January 2019 to 31 December 2020 was used.

After univariable analysis of each putative risk factor, dog age and bodyweight at adoption were selected for use in subsequent modelling as several studies have identified that age and size (or bodyweight) are associated with relinquishment [8,11,33,34] and because the point estimates of hazard ratios from univariable analyses indicated strong associations between these variables and readmission.

Other variables with low *p*-values on univariable analysis were then reassessed, now adjusted for dog age and bodyweight (i.e., each model had three exposure variables: the variable of interest, dog age, and dog bodyweight). Based on results from the trivariable models, there was evidence that six variables were associated with readmission, and two final multivariable models were fitted using these six variables.

In a separate model, hazards of readmission after the last day of the major COVID-lockdown period (27 June 2020) were compared to hazards on or before that date by fitting the corresponding binary indicator variable as a time-varying categorical covariate, with bodyweight at adoption and the four additional variables listed above fitted as covariates.

Some coat colours were not represented in some breeds, and there were few or no adoptions for a considerable number of colour-breed combinations. Accordingly, effects of coat colour and breed were also explored with a further model using only colours and breeds with at least 20 adoptions in each combination of colour and breed. For coat colour, categories with less than 100 adoptions were pooled in these two final models.

When comparing exposure categories (e.g., 10 to <25 kg bodyweight at adoption compared to <10 kg) using Cox models, one key assumption is that the hazards are proportional. This means that the ratio of the log (hazard) of readmission over time for 10 to <25 kg dogs at adoption compared to that for <10 kg dogs is assumed to be constant over the range of times analysed. The validity of this assumption was tested globally, and for each covariate for both models based on Schoenfeld residuals using Stata’s -estat phtest- command.

For each of dog age at adoption, coat colour, days in foster care before adoption, bodyweight at adoption, and breed, the population impact (i.e., the effect of that factor on the overall proportion of adoptions where the dog was readmitted by 90 days for the population) was estimated as:$$\overset{n}{\underset{i=1}{\sum }}FFD_{i}\times P_{i}$$
where: *n* is the number of categories of the variable, *FFD_i_* is the failure function difference for the *i*th category, i.e., the failure function point estimate at 90 days for the *i*th category minus that for the reference category, and *P_i_* is the proportion of adoptions that were in that category. Results are explained in Section 4. The category with lowest hazard was used as the reference category for these calculations. Failure function point estimates were calculated using Stata’s -stcurv- command, with adoption year/whether adoption was in a COVID-lockdown period set to 2020 but not in lockdown, and all other covariates set to their mean values.

## 3. Results

### 3.1. Description of Study Population

There were 6212 dog adoptions from 5587 dogs in the two-year study period (3640 and 2572 in 2019 and 2020, respectively). In total, 5043 dogs were adopted once, 472 twice, 65 three times, 5 four times and 2 five times in the two-year study period.

Just over half of the of the 6212 adoptions (3133) were from the two Brisbane shelters (Wacol (*n* = 2209) and Dakabin (*n* = 924)), with a further 2853 (46%) adopted from the RSPCA’s nine regional shelters, while the remaining 226 (4%) adoptions occurred from three offsite adoption events (See Appendix A
Table A1: Dog adoption numbers per site).

### 3.2. Readmissions

For 865 of the 6212 adoptions, the dog was readmitted during the study follow-up period. Of those 865 readmissions, for 552 (64%), the dog was returned within the first 14 days of their adoption period, and for 119, the dog was returned on the same day as when it was adopted (day 1). Figure 1 illustrates the distribution of readmissions in the first 21 days. These represent 68% of the 865 readmissions with the remaining 32% returned from days 22 to 732. The distribution table for the 865 dogs readmitted during the study period is provided in Appendix A
Table A2. Distribution of the readmittance of the 865 dogs adopted from RSPCA Queensland in 2019 and 2020.

The Kaplan–Meier failure function for readmission of adopted dogs to RSPCA Queensland by day of adoption period is shown in Figure 2a. The adoption failure function increased rapidly in the first 30 days before stabilizing at just under 15%. The Kaplan–Meier smoothed hazard estimates are shown in Figure 2b. What stands out in this graph is that the hazards of readmission were highest in the first 14 days of the adoption period, including being extremely high on days 1–3 of the adoption period, with a short sharp increase around day 14 then a steady decline.

### 3.3. Sources and Reasons for Readmission

Of the 865 readmitted adoptions, the dogs’ owners returned 89% (773). The remaining 11% (92) were readmitted as strays (87) or dogs seized or surrendered after an investigation by a humane officer (5). Five per cent of the owners requested euthanasia (41/773), with 28 of these also giving consent to rehome if RSPCA Queensland determined them suitable to rehome.

The median day of the adoption period when the dog was readmitted for the 773 adoptions where the dog was returned by its owner was day 7 (range days 1 to 691; 25th and 75th percentiles days 2 and 26). For the 87 adoptions where the dog was readmitted as a stray, the median day of the adoption period when the dog was readmitted was day 140 (range days 1 to 732; 25th and 75th percentiles days 33 and 327).

### 3.4. Univariable Analyses

The magnitudes of effects were estimated with hazard ratios. If the HR is known to be 1, there is no difference in the risk of readmission between that and the reference category. If the HR is known to be less than 1 the risk of readmission is lower than adoptions in the refence category. Where the HR is known to be greater than 1 this indicates a higher risk of readmission. From univariable analyses, eight exposure variables had high *p*-values for associations with time to readmission. Descriptive statistics and hazard ratio estimates for these are shown in Table 1. These variables were not included in the subsequent multivariable models.

Table 2 shows results for variables with low *p*-values on univariable analyses of associations with time to readmission. There was evidence that hazard of readmission was increased with: the dog’s second adoption in the study period (2019 and 2020; relative to its first adoption), some coat colours, heavier dogs, some breeds, dogs that were aged ≥6 months, dogs sourced as council admissions, i.e., where the dog is brought in by a local government animal management officer, dogs admitted after being relinquished by owners because they were destructive, dogs that had prior exits from RSPCA Queensland that were not adoptions, dogs that had been temporarily classified as disposition under final review (i.e., reviewed for euthanasia due to serious behavioural attributes), dogs that had no foster care, dogs that had been available for adoption for 30 or more days, and dogs adopted in 2020 during the COVID lockdown. Being univariable analyses, some of these variables will be cofounding others, for example the effect of body weight will have been confounded by age.

The dog’s adoption number and the reason for owner relinquishing the dog prior to the study adoption were not analysed further. Instead, only each dog’s first adoption was used in all other analyses, and reasons for owners relinquishing were not analysed further because most admissions preceding adoptions were not by the owner relinquishing during the study period.

### 3.5. Trivariable Analyses

Associations for other variables in Table 2 were then explored adjusted for dog age and bodyweight, to remove any confounding due to these latter two variables. As effects of foster care appeared similar for various durations of time in foster care (Table 2) this was collapsed to a binary variable (no time or any time in foster care) for further analyses. Results from these trivariable models were also consistent with hazard of readmission being increased with some coat colours, some breeds, dogs that had no foster care, and dogs adopted in 2020 during the COVID lockdown (overall *p*-values for variable: 0.051, 0.017, 0.001, and 0.004, respectively). Dogs that were aged ≥6 months and heavier dogs were also at higher risk (overall *p*-values for age and bodyweight adjusted for the other: 0.014 and <0.001, respectively). These six variables were further explored in final multivariable analyses (See 3.6).

The point estimate for dogs that had been temporarily classified as disposition under final review (adjusted hazard ratio 1.07; 95% CI 0.81 to 1.40; *p* = 0.652) was weaker (i.e., closer to 1.00) than that from univariable analysis (1.37; Table 2). The other two RSPCA behaviour intervention variables (under behaviour management and behaviour consultation needed) were also assessed in trivariable models. Estimates for these (0.88; 95% CI 0.61 to 1.25; *p* = 0.459 and 0.83; 0.51 to 1.38; *p* = 0.477, respectively) were similar to those from the univariable analyses (1.11 and 1.01, respectively; Table 1). 

### 3.6. Final Multivariable Analyses

Two multivariable models were developed with six exposure variables: bodyweight at adoption or breed, dog age at adoption, coat colour, days in foster care before adoption, and adoption year/whether adoption was in a COVID-lockdown period. Bodyweight at adoption and breed were closely correlated so it was inappropriate to fit these variables simultaneously in the same model. Therefore, two multivariable models were fitted, each with one of bodyweight at adoption or breed, and for both models, the remaining four variables: dog age at adoption, coat colour, days in foster care before adoption, and adoption year/whether adoption was in a COVID-lockdown period.

Results for the two multivariable models are shown in Table 3 and Table 4. There was no evidence that the proportional-hazards assumption was violated in these models. Global *p*-values to test this assumption based on Schoenfeld residuals were 0.848 and 0.752 for the models with body weight and breed, respectively, and for both models, *p*-values for individual covariates were >0.06 (most 0.3 to 0.9).

For age at adoption, for dogs <4 months, estimated hazard ratios ranged from 0.53 to 0.69 and *p*-values were 0.013 to 0.063. The results also suggest that the hazard of readmission for dogs that have spent time in foster is less than that for dogs who have spent no time in foster (HR = 0.76, 95% C.I. 0.61 to 0.95).

In addition, there was no evidence of immediate increases in numbers of readmissions when COVID lockdown periods ended. All shelters closed from 18 March to 27 June 2020. Of the 5013 adopted dogs not readmitted by 14 days before the end of that lockdown period, 10 were readmitted in that 14-day period, and 9 were readmitted in the 14-day period commencing the day after the lockdown ended. Wacol, Dakabin, Ipswich, Toowoomba, Kingaroy, and Noosa shelters also closed from 27 August to 19 September 2020. Of the 3605 dogs adopted from those shelters and not readmitted to any RSPCA Queensland shelter by 14 days before the end of that lockdown period, 12 were readmitted in that 14-day period, and 9 were readmitted in the 14-day period commencing the day after the lockdown ended. Hazards of readmission after 27 June 2020 (the last day of the major COVID-lockdown period) were compared to those on or before 27 June 2020. The estimated hazard ratio was 0.93 (95% CI 0.63 to 1.39; *p* = 0.735). These results are not consistent with there being a large change in hazard of readmission between the two periods; however, they do not preclude modest differences in either direction.

As Table 4 illustrates, small toy cross dogs (adjusted HR = 0.60, 95% CI 0.44 to 0.84) and designer poodle cross dogs (adjusted HR = 0.28, 95% CI 0.09 to 0.88) are at reduced risk of readmission when compared to dogs in the working dog cross category. Estimated hazards of readmission were similar for giant dogs (HR = 1.14, 95% CI 0.85 to 1.53) and bull terrier cross dogs (HR = 1.14, 95% CI 0.89 to 1.46) relative to dogs in the working dog cross category.

Estimated effects for the other variables included in the model (age at adoption, coat colour, days in foster care before adoption and adoption year/in COVID-lockdown period) were similar to those effects reported in Table 3.

Using only colours and breeds with at least 20 adoptions in each combination (*n* = 3576 adoptions), numbers of adoptions used, and hazard ratios are shown in Table 5 and Table 6, respectively. The likelihood ratio test *p*-value for terms for interaction between colour and breed was 0.48 so just main effects were fitted. After adjusting for coat colour, hazard ratio estimates still indicate that relative to working dog cross, small toy cross (Chihuahua/Maltese/Terrier) were at reduced risk of readmission (estimated HR = 0.48), and after adjusting for breed, estimates still indicated that brindle dogs and white dogs were at increased risk of readmission (estimated HR = 1.44 and 1.69, respectively) relative to black dogs.

To demonstrate differences in hazard of readmission between covariate patterns, failure functions were generated for a low and a high-risk example adoption. These were estimated using the same model that was used to generate the results reported in Table 3. In Figure 3, the low-risk example (lower line; blue) is for adoptions of black, small toy cross dogs (Chihuahua/Maltese/Terrier) aged 2 to <3 months, who had been in foster care before adoption. The high-risk example (upper line; maroon) is for adoptions of brindle giant breed cross (Wolfhound/Mastiff/Dane) dogs aged 6 to <12 months, who had not been in foster care before adoption. Both examples were for adoptions in 2020 not during lockdown.

Estimates of the maximum impact of strategies to reduce the risk of readmission were calculated for bodyweight and age at adoption, coat colour, spending time in foster care before adoption, and breed using population attributable 90-day failure estimates. These estimate the expected reduction in the cumulative percentage of dogs readmitted by day 90 (including the high-risk period from days 1 to 21) if the hazards of readmission for higher risk categories were reduced to those of a lower risk category. They are thus estimated maximum impact of strategies, impacts that would be achieved only if the strategy was fully successful in reducing hazards to those of a lower risk category.

Results are in Table 7. The overall cumulative percentage of study dogs readmitted by the 90th day of the adoption period was 10.8%. Expected reductions for individual factors ranged from 1.8% to 3.6% with one additional estimate of 6.8%. If the hazards of readmission for dogs weighing ≥10 kg were reduced to those of dogs weighing <10 kg, the overall cumulative percentage of dogs readmitted is estimated as decreasing by 2.0 percentage points (i.e., from 10.8% to 8.8%). Were hazards of readmission for dogs aged ≥2 months reduced to those for dogs 1 to <2 months, the overall cumulative percentage of dogs readmitted is estimated as decreasing by 3.6 percentage points. For non-black dogs, if hazards were reduced to those for black dogs, the expected decrease is 2.1 percentage points, and for dogs not receiving foster care, if hazards were reduced to those for dogs that had foster care, the expected decrease would be 1.8 percentage points. Finally, if the hazards of readmission for all breeds other than for designer poodle cross dogs were reduced to those of small toy cross (Chihuahua/Maltese/Terrier), the expected decrease would be 3.5 percentage points.

## 4. Discussion

This research aimed to identify risk factors which influence the hazards of readmission (and hence time to readmission) for dogs adopted from RSPCA Queensland shelters. At the time of writing, we were not aware of other studies using survival analysis to investigate time-to-readmission post adoption. Survival analysis provides a unique insight into the possible relationships between variables and the impact these can have on the time to the event being studied; for example, the time it takes for a dog in a shelter to be adopted or for an adopted dog to be returned post-adoption.

Several adoption studies have used survival analysis; however, these studies investigated risk factors associated with the time-to-adoption for dogs in shelters. Diesel et al. [35] found several variables affected time-to-adoption, many of which such as size, breed, age, and coat colour, recur in research on adoptability. Cain et al. [29] suggest that phenotypic characteristics are associated with the hazard of adoption which is also influenced by their time in the shelter. Kay et al. [26] used a similar approach to determine the effects of shelter and animal characteristics on length of stay in the shelter while Patronek and Crowe [25] used survival analysis to explore factors associated with live release rates for dogs in a large US shelter.

In our study, the long-term percentage for readmission after adoption was just under 15%. The daily adoption ‘failure’ rates (i.e., the hazards of readmission) was highest in the first 30 days of the adoption period, followed by a gradual decline. The highest hazards of readmission occurred within the first 14 days of the adoption. Just under 64% of readmitted dogs were returned in that period. The spike in returns around day 14 could be attributed to the RSPCA Qld policy to refund adoption fees up to and including 14 days post-adoption. The first month post adoption is recognized as a critical time of transition for both the dogs and their adopters. Therefore, could offering a refund at this time negatively impact retention. It may be useful to investigate extending the refund period in combination with providing more tailored support for the higher risk groups or reviewing the overall efficacy of a such a policy.

A striking feature of the readmission pattern was the number of dogs returned on the day of their adoption or the next day (days 1 and 2, respectively). On the first day of their adoption, 119 dogs (14% of total readmissions) came back. An important question from the study is, what happened in that first day to motivate 119 people to return their dog so quickly? Moreover, what could be done to improve retention? Shore [16] found that several adopters observed problematic behaviour within the first 24 h, leading them to return the dog, although it may have taken a few weeks for them to relinquish it. Rescue organisations are acutely aware of the need for post-adoption support. Many employ a range of activities, including follow-up phone calls, printed and digital information, and in some cases, dog/owner counselling, online resources, and training. Studies confirm that getting uptake or engagement with these resources from adopters can be challenging [14,36,37]. Hawes et al. [10] questioned the efficacy of preventive education programs, suggesting that many of these ignore the systemic issues that may be of greater influence, for example, access to transport and affordable veterinary care. Similarly, Protopopova and Gunter [38], in their review of shelter intervention programs aimed at reducing relinquishment, suggest that many ‘one-size-fits-all’ programs have met with mixed success. They report a lack of research into programs that would impact factors leading to relinquishment, many of which, it is speculated, do not relate to the dog. Thus, further research is needed to identify factors influencing adoption retention and success. Understanding both adopter and dog factors is fundamental to developing and tailoring programs that target these factors, as well as determining the best time and way to deliver these [22,39,40,41]. Accompanying this work is the need to consider how to support adopters in their choice of dogs, taking into consideration its needs and how these complement the lifestyle, capability, and capacity of the adopter [14,42,43].

The beneficial effects of dogs spending time in foster care before adoption is a promising finding; foster time was associated with a lower risk of readmission. This is concordant with several studies that have examined the positive impact of foster care on the outcomes for dogs in shelter organisations [15,44,45,46,47]. Foster care appears to provide a preparatory basis for the transition to a home environment. Using foster care also frees up shelter space and reduces shelter staff hours and costs. This enables the reallocation of often stretched resources. Some shelters have also introduced differing foster levels and invested more time working with foster carers to build numbers and capacity. In some instances, foster carers are paid to take on the more challenging cases [48]. In the current study, times in foster care varied substantially. Foster care for the study dogs was typically more than 6 days, and in some instances extended to 84 days or more. From our univariable results, hazards of readmission appeared similar for dogs adopted after shorter and longer periods in foster care, and even a foster care period of 7 to 13 days appeared beneficial. However, these results should not be interpreted as indicating that 7 to 13 days is a sufficient duration for all dogs. In the study population, dogs in foster care were monitored for suitability for adoption and not adopted until considered suitable. Thus, times in foster care, in part, reflect the times taken for dogs to become suitable, and longer times in foster care have additional benefits over shorter times for some dogs. The marginal benefits of additional time in foster care are also obscured in the current study because some dogs that became suitable for adoption remained in foster care until selected or until legal proceedings were completed.

Our study also identified body weight at adoption as a risk factor associated with time to readmission. Hazard ratios for the three heavier bodyweight categories indicated that these dogs are at increased risk of readmission compared with the reference category (<10 kg). It is interesting to note that weight and size also recur as factors associated with an increased length of stay before adoption, and a reason given for returning an adopted dog [33,49,50,51].

Lifestyle and living situations might influence adopter preference concerning weight/size; for example, many rental properties and medium to high-density apartment blocks and retirement complexes stipulate the weight of dogs allowed to be kept. The stipulated weight is often not more than 10 kg. Additionally, affordability of care and management of a larger dog may influence adopter choice. Although the number of larger and giant dogs is proportionately lower in this study population, it does highlight a potential need to tailor adoption processes to ensure the best fit with an owner and the suitability of the home environment as well as appropriate post-adoption support to upskill owners to care for, train and manage their new dog [22,52]. This highlights a potential need to tailor resources and work with adopters to ensure the best fit of dog with both the owner and their home environment including appropriate post-adoption support [52,53].

From our findings, young dogs (<3–4 months) at adoption are less likely to be readmitted compared to dogs 6 months or older at adoption. Where a dog is beyond the puppy stage—perhaps poorly socialized, had little or no training, and/or their owners have little knowledge about dogs’ needs and behaviours—it could be more difficult for some owners to manage and retain these dogs. In looking across reported findings on age, there is agreement and contradiction, driven partly by differences in scope and methodology, and potentially by location and culture. Powell et al. [54] in a study on returns found that dogs older than 6 months, and medium to large size, at adoption, were more likely to be returned. Cain et al. [29] found that when comparing senior to adult dogs, senior dogs had a lower chance of adoption as the length of stay increased until the 50-day mark, when senior dogs were more likely to be adopted than adult dogs. They suggested this finding may have been influenced by adopters seeking to adopt senior and end-of-life stage dogs, in conjunction with effective communication strategies promoting senior dogs.

Coat colour is recognized as a key characteristic influencing the chance of a dog being adopted/time to adoption [35,49,55,56]. Our findings showed that coat colour is also related to risk of readmission. Brindle dogs, tan dogs and white dogs were at increased risk of readmission when compared to black dogs. Associations between coat colour and risk of readmission would be confounded by breed and vice versa, given that many breeds do not exhibit the full range of coat colours. Breeds with brindle, white, or tan coats include the Bull Terriers, Mastiffs, Great Danes, Boxers, and Greyhounds. However, after adjusting for coat colour, hazard ratio estimates still indicated that relative to working dog cross, small toy cross (Chihuahua/Maltese/Terrier) were at reduced risk of readmission, and after adjusting for breed, estimates still indicated that brindle dogs and white dogs were at increased risk of readmission relative to black dogs. It is acknowledged that precise identification of breed is problematic, noting that most of dogs in this study were of mixed breeds. In designating their breed labels on their initial shelter admission, which were based largely on their morphology, some level of misclassification was likely.

Several research studies have examined the impact of breed labelling on adopter choice and the related time in foster. Cohen et al. [57] found that dogs’ median length of stay in a US shelter decreased by 37% once breed labels were removed. Gunter [15] argues that breed identification and labelling are of questionable value due to the unreliability of identification of shelter dog breed heritage. Breed identification, coupled with public perception of breed stereotypes, could negatively impact the adoptability of shelter dogs. She also suggests that shelters would increase adoptions if they focused on promoting the morphology and behaviour of dogs to support adopters in making more informed choices, where the match might be more suited to lifestyle and the home environment.

Since the COVID-19 pandemic outbreak, there has been an increasing interest in, and debate over the fall-out for so-called COVID puppies post lockdown and the lifting of mandated work-from-home directives. Some have speculated that shelters would face an influx of relinquished pets once people returned to pre-COVID work–life patterns whereby they no longer had the time to care for a pet and potentially the incidence of dog behavioural issues such as separation anxiety would increase and may influence decisions to relinquish the pet [58,59,60]. Our findings, however, showed no evidence of immediate increases in the hazard of readmission when COVID lockdown periods ended. This could be influenced by many people continuing to working from home, on a full or part-time basis; however, lockdown periods varied by shelter so this lack of association might have been confounded by the geographic location of the adopter.

In the final analyses, population impact estimates were calculated for bodyweight and age at adoption, coat colour, spending time in foster care before adoption, and breed. Findings indicated that if hazards for readmission for all categories reduce to the lowest risk levels, the percentage of adoptions where dogs return could be reduced by 2.1 to 6.8%, only if the intervention strategy is wholly successful. In practical terms, shelters cannot alter the inherent characteristics of their dogs and understanding the relationship to readmission for white dogs, or giant dogs is complex and possibly confounded by broader people factors. These population impact findings serve mainly to invoke caution in balancing potential outcomes for an individual or a subset of dogs and the related investment of resources to achieve a change. For example, in this study the older, larger dogs may be more challenging to rehome and at a higher risk of readmission. However, there is a small number of these dogs; therefore, should resources be allocated to promote these dogs specifically, or should a foster-to-adoption program with a tailored support component be provided for all categories dogs in care? Alternatively, is it accepted that these dogs might take longer to find a long-term home and that a specific intervention is not cost-effective for the potential reduction of their risk of readmission?

## 5. Limitations

Limitations of the present study include the potential breadth and variability of categories and subcategories in recording colour and breed, and the integrity and consistency of data entry. Additionally, we could monitor only for readmissions to the RSPCA Queensland, and not all dogs adopted from RSPCA shelters that are surrendered post adoption are returned to an RSPCA Queensland shelter. Caution is advised in generalizing from these results as they are based on the shelter dogs specific to this study cohort and as such these results might not be wholly applicable to different dog populations or different shelters. It is noted that the demographics of shelter animals is dynamic and influenced by many factors. However, the positive impact of foster, and the usefulness of knowing the prospective adoption population to enable more tailored support or resource allocation could support improved retention of adopted dogs.

## 6. Conclusions

Not all adoptions are successful for some dogs the first, second or third time around. In our study approximately two-thirds of all returns occurred during the first 14 days of their adoption period. The cumulative percentage of readmitted adoptions plateaued at just under 15%. Dog size, age, coat colour, breed, and spending time in foster before adoption were identified as risk factors for readmission. Failure functions for a low and a high-risk adoption example demonstrate the large degree of difference in hazard of readmission between covariate patterns. Adoptions of black, small toy cross dogs aged 2 to <3 months, who had been in foster care before adoption were at low risk of return, with an estimated 2% being returned by 90 days, for example. In contrast, adoptions of brindle giant breed cross dogs aged 6 to <12 months, who had not been in foster care before adoption, were much more likely to be returned after adoption, with an estimated 17% being returned by 90 days.

Our findings support the growing interest in foster care’s potential to improve adoption outcomes and reduce the risk of readmission for dogs. Time in foster appears to support a more successful transition to the new home environment. Behaviour support provided for dogs during foster care may also improve their outcomes. These findings highlighted the profile of the higher-risk dogs, potentially providing shelters with an opportunity to examine where and how resources could be allocated to maximize outcomes for the overall cohort.

Our picture of readmission is incomplete. It would be essential to learn more about the people who adopted during the study period—those who kept their dogs and those who returned them to RSPCA Queensland. Gaining insight about the transition to home in the first 24 h to 14 days could be invaluable in understanding how to improve retention in the home and reduce the risk of readmission.

Exploring adopters’ experiences would provide a more in-depth insight into their adoption journey and the risks contributing to readmission. These could highlight possible gaps in knowledge about the dog population, adoption processes and the support provided to adopters. Successful adoptions deliver benefits for the adopter and the dog. This success also benefits rescue organization and community animal management agencies by reducing the impost on human and financial resources.

## Figures and Tables

**Figure 1 animals-12-02568-f001:**
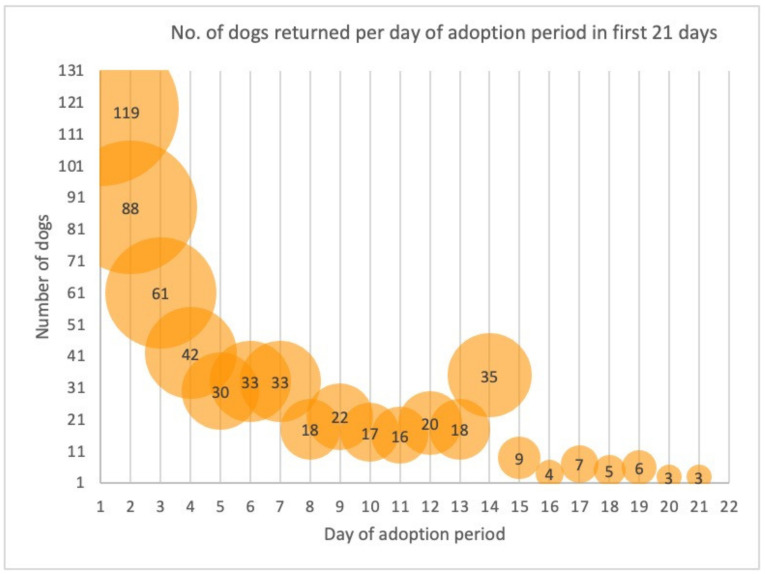
Distribution of the numbers of adopted dogs that were readmitted to RSPCA Queensland in 2019 and 2020 during for the first 21 days of their adoption period; day of adoption is day 1; 119 dogs were readmitted on the day of their adoption, 88 on the next day (the second day of their adoption period), and so on.

**Figure 2 animals-12-02568-f002:**
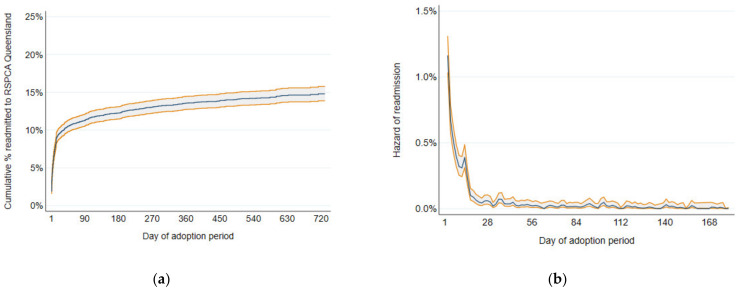
(**a**) Kaplan–Meier failure function for readmission to RSPCA Queensland of adopted dogs by day of adoption period; (**b**) Kaplan–Meier smoothed hazard estimates for readmission to RSPCA Queensland for dogs adopted from RSPCA Queensland in 2019 and 2020, by day of adoption period. Shaded bands indicate pointwise 95% confidence intervals.

**Figure 3 animals-12-02568-f003:**
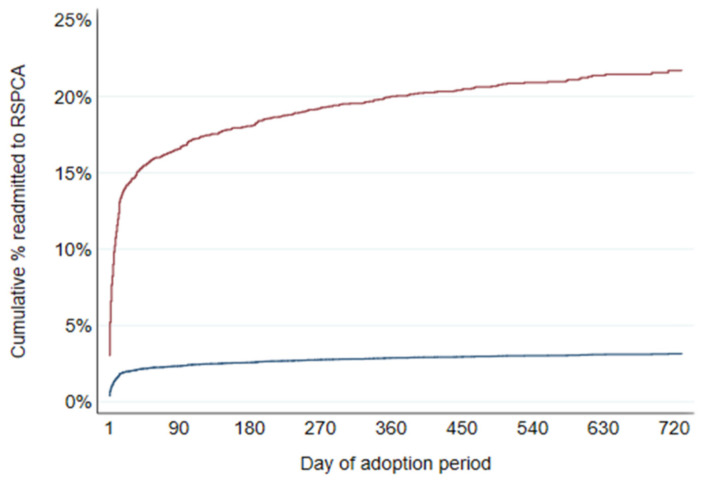
Failure functions for readmission to RSPCA Queensland of dogs adopted from RSPCA Queensland in 2019 and 2020, by day of adoption period for a high-risk example (upper line; maroon) and a low-risk example (lower line; blue) example covariate pattern. Population impacts.

**Table 1 animals-12-02568-t001:** Descriptive statistics and hazard ratio estimates for exposure variables with high overall *p*-values on univariable analyses of associations with time to readmission for dogs adopted from RSPCA Queensland in 2019 and 2020.

Factor and Level	No. Adoptions ^1^	No. Readmitted ^1^	% Readmitted	Hazard Ratio	95% CI	*P* ^2^
*Dog’s sex*						**0.377**
Female	2766	385	13.9%	Reference cat.		
Male	2821	369	13.1%	0.94	0.81 to 1.08	0.377
Pooled	5587	754	13.5%			
2. *Dog’s prior number of admissions to RSPCA Queensland (up to and including admission immediately before study adoption) ^2^*						**0.161**
1	5067	668	13.2%	Reference cat.		
2	371	60	16.2%	1.23	0.94 to 1.60	0.132
3 to 8	149	26	17.4%	1.31	0.89 to 1.94	0.177
Pooled	5587	754	13.5%			
3. *Dog’s prior number of adoptions from RSPCA Queensland ^3^*						**0.537**
0	5392	729	13.5%	Reference cat.		
1	167	23	13.8%	0.99	0.65 to 1.50	0.965
2 to 4	28	2	7.1%	0.50	0.12 to 1.99	0.324
Pooled	5587	754	13.5%			
4. *Under behaviour management*						**0.564**
No	5366	722	13.5%	Reference cat.		
Yes	221	32	14.5%	1.11	0.78 to 1.58	0.557
Pooled	5587	754	13.5%			
5. *Behaviour consultation needed*						**0.970**
No	5466	738	13.5%	Reference cat.		
Yes	121	16	13.2%	1.01	0.62 to 1.66	0.970
Pooled	5587	754	13.5%			
6. *Adopted off site*						**0.981**
No	5257	708	13.5%	Reference cat.		
Yes	330	46	13.9%	1.00	0.74 to 1.35	0.980
Pooled	5587	754	13.5%			
7. *Month adopted*						**0.228**
Jan	710	101	14.2%	Reference cat.		
Feb	666	104	15.6%	1.11	0.85 to 1.46	0.444
Mar	500	52	10.4%	0.73	0.52 to 1.02	0.068
Apr	374	47	12.6%	0.88	0.62 to 1.25	0.473
May	413	67	16.2%	1.19	0.87 to 1.62	0.279
Jun	406	56	13.8%	1.00	0.72 to 1.39	0.987
Jul	477	58	12.2%	0.89	0.64 to 1.23	0.474
Aug	407	59	14.5%	1.08	0.78 to 1.49	0.645
Sep	384	55	14.3%	1.07	0.77 to 1.49	0.684
Oct	406	60	14.8%	1.13	0.82 to 1.56	0.452
Nov	387	40	10.3%	0.79	0.55 to 1.14	0.215
Dec	457	55	12.0%	0.94	0.67 to 1.30	0.693
Pooled	5587	754	13.5%			
8. *Adopter’s postcode’s index of social disadvantage (decile) ^4^*						**0.349**
1	374	54	14.4%	Reference cat.		
2	777	127	16.3%	1.14	0.83 to 1.56	0.432
3	360	49	13.6%	0.95	0.64 to 1.39	0.776
4	446	66	14.8%	1.05	0.74 to 1.51	0.777
5	737	98	13.3%	0.93	0.66 to 1.29	0.654
6	792	107	13.5%	0.94	0.68 to 1.31	0.717
7	545	61	11.2%	0.77	0.54 to 1.12	0.170
8	726	93	12.8%	0.89	0.64 to 1.25	0.513
9	438	49	11.2%	0.78	0.53 to 1.15	0.205
10	368	47	12.8%	0.89	0.60 to 1.32	0.575
Not available	24	3	12.5%			
Pooled	5587	754	13.5%			

^1^ For analyses of all putative risk factors other than dog’s adoption number in 2019 or 2020, only the dog’s first admissions in that period were used. ^2^ Bolded values are overall univariable likelihood ratio test *p*-values for variables; unemboldened values are Wald *p*-values for each category relative to reference category ‘‘Reference cat’’) from univariable analysis. ^3^ Numbers of admissions, exits and adoptions on or since 1 January 2011 and prior to the study adoption. ^4^ 1 indicates adopter’s postcode area had high relative social disadvantage (lowest decile of scores by area nationally) and 10 indicates adopter’s postcode area had low relative social disadvantage (highest decile of scores).

**Table 2 animals-12-02568-t002:** Descriptive statistics and hazard ratio estimates for exposure variables with low overall *p*-values on univariable analyses of associations with time to readmission for dogs adopted from RSPCA Queensland in 2019 and 2020.

Factor and Level	No. Adoptions ^1^	No. Readmitted ^1^	% Readmitted	Hazard Ratio	95% CI	*P* ^2^
*Dog’s adoption number in 2019 or 2020*						**0.010**
First	5587	754	13.5%	Reference cat.		
Second	544	98	18.0%	1.37	1.11 to 1.69	0.003
Third	72	11	15.3%	1.23	0.70 to 2.15	0.481
Fourth	7	2	28.6%			
Fifth	2	0	0.0%			
Pooled	6212	865	13.9%			
2. *Coat colour*						**0.015**
Black	1318	137	10.4%	Reference cat.		
Brindle	1025	167	16.3%	1.60	1.27 to 2.00	<0.001
Chocolate	118	16	13.6%	1.34	0.80 to 2.25	0.265
Fawn	130	22	16.9%	1.66	1.06 to 2.60	0.028
Merle	139	23	16.5%	1.61	1.04 to 2.51	0.034
Red	163	19	11.7%	1.12	0.69 to 1.81	0.641
Roan/blue/red	181	22	12.2%	1.18	0.75 to 1.85	0.475
Sable/sadle	159	18	11.3%	1.10	0.67 to 1.80	0.705
Tan	767	105	13.7%	1.33	1.04 to 1.72	0.026
Tricolour	156	22	14.1%	1.08	0.71 to 1.63	0.714
White	87	5	5.7%	1.46	1.15 to 1.85	0.002
Other	425	61	14.4%	1.40	1.04 to 1.90	0.028
Pooled	5587	754	13.5%			
3. *Bodyweight at adoption (kg)*						**<0.001**
<10	2060	176	8.5%	Reference cat.		
10 to <25	2129	322	15.1%	1.83	1.52 to 2.20	<0.001
25 to <45	1230	227	18.5%	2.28	1.87 to 2.77	<0.001
≥45	63	16	25.4%	3.30	1.98 to 5.51	<0.001
Not recorded	105	13	12.4%			
Pooled	5587	754	13.5%			
4. *Predominant breed label (morphologically)*						**0.015**
Working dog cross (Kelpie/Cattle Dog/Koolie)	906	118	13.0%	Reference cat.		
Australian hybrid cross (Bull Arab/Bullhound)	589	77	13.1%	1.00	0.75 to 1.33	0.997
Border Collie cross	312	31	9.9%	0.76	0.51 to 1.13	0.170
Bull Terrier cross	1327	208	15.7%	1.22	0.98 to 1.53	0.081
Bulldog cross	109	14	12.8%	0.99	0.57 to 1.72	0.959
Designer poodle cross	59	3	5.1%	0.37	0.12 to 1.17	0.092
Giant breed cross (Wolfhound/Mastiff/Dane)	527	85	16.1%	1.26	0.95 to 1.66	0.108
Large breed cross (German Shepherd/Ridgeback)	712	93	13.1%	1.00	0.76 to 1.31	0.995
Medium breed cross (Husky/Malamute/Vizsla)	306	43	14.1%	1.08	0.76 to 1.53	0.671
Purebred (wide range of breeds)	153	22	14.4%	1.11	0.70 to 1.75	0.657
Small toy cross (Chihuahua/Maltese/Terrier)	587	60	10.2%	0.78	0.57 to 1.07	0.123
Pooled	5587	754	13.5%			
5. *Age at adoption (months)*						**<0.001**
<2	201	12	6.0%	0.34	0.19 to 0.62	<0.001
2 to <3	929	66	7.1%	0.41	0.30 to 0.55	<0.001
3 to <4	438	40	9.1%	0.53	0.37 to 0.76	<0.001
4 to <6	306	36	11.8%	0.69	0.48 to 1.00	0.051
6 to <12	784	131	16.7%	Reference cat.		
12 to <18	356	68	19.1%	0.98	0.75 to 1.28	0.879
18 to <24	545	89	16.3%	1.17	0.87 to 1.56	0.305
24 to <36	553	91	16.5%	1.00	0.77 to 1.31	0.994
36 to <48	368	47	12.8%	0.76	0.54 to 1.05	0.099
48 to <60	275	47	17.1%	1.04	0.75 to 1.46	0.801
60 to <72	227	36	15.9%	0.97	0.67 to 1.41	0.883
72 to <84	185	30	16.2%	0.98	0.66 to 1.45	0.910
84 to <96	133	23	17.3%	1.04	0.67 to 1.62	0.866
≥96	280	38	13.6%	0.83	0.58 to 1.19	0.304
Not recorded	7	0	0.0%			
Pooled	5587	754	13.5%			
6. *Source of dog for admission immediately before study adoption*						**0.001**
Owner surrender	1507	195	12.9%	Reference cat.		
Ambulance	106	10	9.4%	0.73	0.39 to 1.38	0.333
Council	1273	220	17.3%	1.36	1.12 to 1.65	0.002
Humane Officer	950	109	11.5%	0.89	0.70 to 1.13	0.334
Offspring	144	10	6.9%	0.53	0.28 to 1.00	0.052
Return	804	108	13.4%	1.36	0.51 to 3.67	0.539
Stray	22	4	18.2%	1.04	0.82 to 1.31	0.755
Transfer In	766	98	12.8%	0.98	0.77 to 1.25	0.887
Not recorded	15	0	0.0%			
Pooled	5587	754	13.5%			
7. *Reason for owner relinquishing dog at admission preceding study adoption*						**0.004**
Owner/owner’s circumstances	442	71	16.1%	Reference cat.		
Aggression	22	5	22.7%	1.44	0.58 to 3.57	0.430
Destructive	10	5	50.0%	3.40	1.37 to 8.41	0.008
Escaping	64	7	10.9%	0.66	0.30 to 1.44	0.295
Behaviour other	23	4	17.4%	1.06	0.39 to 2.90	0.909
Expectation mismatch-too big, too boisterous, no time	73	9	12.3%	0.77	0.38 to 1.54	0.459
Expense	165	22	13.3%	0.83	0.52 to 1.35	0.459
Incompatible with household member or another pet	44	8	18.2%	1.11	0.53 to 2.30	0.786
Poor choice	59	6	10.2%	0.60	0.26 to 1.37	0.223
Other	471	40	8.5%	0.50	0.34 to 0.73	<0.001
Reason not recorded	156	22	14.1%			
Not admitted by owner relinquishment	4058	555	13.7%			
Pooled	5587	754	13.5%			
8. *Dog had prior exits from RSPCA Queensland that were not adoptions ^3^*						**0.015**
No	5235	690	13.2%	Reference cat.		
Yes	352	64	18.2%	1.40	1.08 to 1.80	0.011
Pooled	5587	754	13.5%			
9. *Disposition under final review*						**0.028**
No	5262	696	13.2%	Reference cat.		
Yes	325	58	17.8%	1.37	1.05 to 1.79	0.021
Pooled	5587	754	13.5%			
10. *Days in foster care before adoption*						**<0.001**
0	4340	648	14.9%	Reference cat.		
1 to <7	107	10	9.3%	0.61	0.33 to 1.14	0.119
7 to <14	222	16	7.2%	0.47	0.28 to 0.76	0.003
14 to <21	221	20	9.0%	0.59	0.38 to 0.92	0.021
21 to <42	320	29	9.1%	0.60	0.41 to 0.87	0.007
42 to <84	246	23	9.3%	0.61	0.40 to 0.92	0.019
≥84	131	8	6.1%	0.41	0.20 to 0.82	0.012
Pooled	5587	754	13.5%			
11. *Days available for adoption*						**0.001**
0 to <1	487	59	12.1%	Reference cat.		
1 to <2	914	108	11.8%	0.98	0.71 to 1.35	0.910
2 to <3	689	90	13.1%	1.10	0.79 to 1.52	0.576
3 to <4	474	50	10.5%	0.87	0.60 to 1.27	0.481
4 to <5	319	40	12.5%	1.05	0.70 to 1.57	0.803
5 to <6	253	30	11.9%	0.99	0.64 to 1.54	0.969
6 to <7	208	34	16.3%	1.42	0.93 to 2.16	0.106
7 to <8	172	25	14.5%	1.25	0.78 to 1.99	0.357
8 to <9	154	26	16.9%	1.45	0.92 to 2.31	0.112
9 to <10	146	20	13.7%	1.14	0.69 to 1.89	0.618
10 to <15	422	72	17.1%	1.46	1.04 to 2.06	0.031
15 to <20	273	41	15.0%	1.28	0.86 to 1.90	0.231
20 to <30	249	39	15.7%	1.30	0.86 to 1.94	0.210
30 to <60	298	61	20.5%	1.76	1.23 to 2.52	0.002
≥60	135	33	24.4%	2.12	1.39 to 3.25	0.001
Not recorded	394	26	6.6%			
Pooled	5587	754	13.5%			
12. *Adoption year/adopted in COVID-lockdown period*						**0.005**
2019 (no lockdown)	3275	493	15.1%	Reference cat.		
2020 not during lockdown	1717	206	12.0%	0.88	0.75 to 1.04	0.128
2020 during lockdown	595	55	9.2%	0.65	0.49 to 0.87	0.003
Pooled	5587	754	13.5%			

^1^ For analyses of all putative risk factors others other than dog’s adoption number in 2019 or 2020, only the dog’s first admissions in that period were used. ^2^ Bolded values are overall univariable likelihood ratio test *p*-values for variables; unemboldened values are Wald *p*-values for each category relative to reference category ‘‘Reference cat’’) from univariable analysis. ^3^ Numbers of admissions, exits and adoptions on or since 1 January 2011 and prior to the study adoption.

**Table 3 animals-12-02568-t003:** Adjusted hazard ratio estimates for associations between putative risk factors and time to readmission to RSPCA Queensland for dogs adopted from RSPCA Queensland in 2019 and 2020. Bodyweight rather than breed was used in this model.

Risk Factor and Level	Adjusted Hazard Ratio ^1^	95% CI	*P* ^2^
*Body weight at adoption (kg)*			**0.001**
<10	Reference cat.		
10 to <25	1.34	1.05 to 1.70	0.020
25 to <45	1.59	1.22 to 2.06	<0.001
≥45	2.20	1.28 to 3.78	0.004
*Age at adoption (months)*			**0.086**
<2	0.53	0.27 to 1.02	0.056
2 to <3	0.62	0.43 to 0.90	0.013
3 to <4	0.69	0.47 to 1.02	0.063
4 to <6	0.77	0.52 to 1.13	0.177
6 to <12	Reference cat.		
≥12	0.94	0.77 to 1.15	0.566
*Coat colour*			**0.078**
Black	Reference cat.		
Brindle	1.52	1.21 to 1.91	<0.001
Chocolate	1.35	0.80 to 2.26	0.263
Fawn	1.56	0.98 to 2.48	0.059
Merle	1.54	0.99 to 2.40	0.056
Red	1.01	0.62 to 1.64	0.964
Roan/blue/red	1.21	0.77 to 1.91	0.399
Sable/sadle	1.22	0.74 to 1.99	0.437
Tan	1.30	1.01 to 1.69	0.045
Tricolour	1.08	0.71 to 1.64	0.711
White	1.41	1.10 to 1.79	0.006
Other	1.34	0.99 to 1.82	0.058
*Foster care before adoption*			**0.015**
No	Reference cat.		
Yes	0.76	0.61 to 0.95	0.017
*Adoption year/in COVID-lockdown period*			**0.013**
2019 (no lockdown)	Reference cat.		
2020 not during lockdown	0.90	0.76 to 1.07	0.226
2020 during lockdown	0.67	0.50 to 0.89	0.006

^1^ Adjusted for the other four variables in this table. ^2^ Bolded values are overall likelihood ratio test *p*-values for variables; unemboldened values are Wald *p*-values for each category relative to reference category ‘‘Reference cat’’).

**Table 4 animals-12-02568-t004:** Adjusted hazard ratio estimates for the association between breed and time-to-readmission to RSPCA Queensland for dogs adopted from RSPCA Queensland in 2019 and 2020. Breed rather than bodyweight was used in this model.

Predominant Breed (Morphologically)	Adjusted Hazard Ratio ^1^	95% CI	*P* ^2^
Working dog cross (Kelpie/Cattle Dog/Koolie)	Reference cat.		
Australian hybrid cross (Bull Arab/Bullhound)	0.93	0.68 to 1.26	0.632
Border Collie cross	0.77	0.51 to 1.16	0.216
Bull Terrier cross	1.14	0.89 to 1.46	0.297
Bulldog cross	0.81	0.46 to 1.43	0.467
Designer poodle cross	0.28	0.09 to 0.88	0.029
Giant breed cross (Wolfhound/Mastiff/Dane)	1.14	0.85 to 1.53	0.395
Large breed cross (German Shepherd/Ridgeback)	0.94	0.71 to 1.25	0.679
Medium breed cross (Husky/Malamute/Vizsla)	1.01	0.70 to 1.44	0.978
Purebred (wide range of breeds)	0.85	0.54 to 1.36	0.503
Small toy cross (Chihuahua/Maltese/Terrier)	0.60	0.44 to 0.84	0.003

^1^ Adjusted for age at adoption, coat colour, days in foster care before adoption and adoption year/in COVID-lockdown period as categorised in Table 3 results for age at adoption, coat colour, foster care before adoption and adoption year/in COVID-lockdown period adjusted for predominant breed were similar to those when adjusted for bodyweight (as reported in Table 3). ^2^ Wald *p*-values for each category relative to reference category (‘Reference cat.’); overall likelihood ratio test *p*-value for breed was 0.001.

**Table 5 animals-12-02568-t005:** Numbers of adoptions by dog coat colour and breed used to assess associations between these variables and time-to-readmission to RSPCA Queensland for dogs adopted from RSPCA Queensland in 2019 and 2020.

Coat Colour	Predominant Breed (Morphologically)							
	Working Dog Cross Kelpie Cattle Dog Koolie	Australian Hybrid Cross Bull Arab Bullhound	Bull Terrier Cross	Giant Breed Cross Wolfhound Mastiff Dane	Large Breed Cross German Shepherd Ridgeback	Medium Breed Cross Husky Malamute Vizsla	Small Toy Cross Chihuahua Maltese Terrier	Pooled
Black	250	73	212	86	262	88	98	1069
Brindle	91	151	398	146	118	35	24	963
Tan	88	57	278	110	110	30	64	737
White	80	221	152	73	28	42	211	807
Pooled	509	502	1040	415	518	195	397	3576

**Table 6 animals-12-02568-t006:** Hazard ratios for dog coat colour and breed adjusted for each other on time-to-readmission to RSPCA Queensland for dogs adopted from RSPCA Queensland in 2019 and 2020.

Breed and Colour ^1^	Adjusted Hazard Ratio ^2^	95% CI	*P* ^3^
*Coat colour*			**0.001**
Black	Reference cat.		
Brindle	1.44	1.12 to 1.87	0.005
Tan	1.24	0.94 to 1.63	0.136
White	1.69	1.29 to 2.21	<0.001
*Breed*			**0.001**
Working dog cross (Kelpie/Cattle Dog/Koolie)	Reference cat.		
Australian hybrid cross (Bull Arab/Bullhound)	0.88	0.62 to 1.25	0.475
Bull Terrier cross	1.05	0.78 to 1.40	0.765
Giant breed cross (Wolfhound/Mastiff/Dane)	1.04	0.74 to 1.47	0.818
Large breed cross (German Shepherd/Ridgeback)	0.88	0.62 to 1.24	0.457
Medium breed cross (Husky/Malamute/Vizsla)	1.11	0.73 to 1.71	0.621
Small toy cross (Chihuahua/Maltese/Terrier)	0.48	0.32 to 0.73	0.001

^1^ Model was fitted using only colours and breeds with at least 20 adoptions in each combination of colour and breed. ^2^ Adjusted for breed and coat colour categorised as shown in this table along with age at adoption, days in foster care before adoption and adoption year/in COVID-lockdown period as categorised in Table 3. ^3^ Bolded values are overall likelihood ratio test *p*-values for variables; unemboldened values are Wald *p*-values for each category relative to reference category ‘‘Reference cat’’).

**Table 7 animals-12-02568-t007:** Population attributable 90-day failure estimates for risk factors for readmission to RSPCA Queensland of dogs adopted from RSPCA Queensland in 2019 and 2020.

Risk Factor and Categories	No. Adoptions	% of All Adoptions	% Readmitted by 90 Days ^1^	Population Attributable 90-Day Failure Estimate
*Body weight at adoption (kg)*				2.0%
<10	2060	38%	7.9%	
10 to <25	2129	39%	10.3%	
25 to <45	1230	22%	12.2%	
≥45	63	1%	16.5%	
Not recorded	105			
Pooled	5587			
*Age at adoption (months)*				3.6% (2.6%) ^2^
<2	201	4%	6.2%	
2 to <3	929	17%	7.3%	
3 to <4	438	8%	8.1%	
4 to <6	306	5%	8.9%	
6 to <12	784	14%	11.5%	
≥12	2922	52%	10.9%	
Not recorded	7			
Pooled	5587			
*Coat colour*				2.1%
Black	1318	24%	7.8%	
Brindle	1025	18%	11.6%	
Chocolate	118	2%	10.3%	
Fawn	130	2%	11.9%	
Merle	139	2%	11.7%	
Red	163	3%	7.9%	
Roan_blue_red_	181	3%	9.4%	
Sable_Sadle	159	3%	9.4%	
Tan	767	14%	10.0%	
Tricolour	243	4%	8.4%	
White	919	16%	10.8%	
Other	425	8%	10.3%	
Pooled	5587			
*Foster care before adoption*				1.8%
No	4340	78%	10.3%	
Yes	1247	22%	8.0%	
Pooled	5587			
*Breed*				3.5% (6.8%) ^3^
Australian hybrid cross (Bull Arab/Bullhound)	589	10.5%	9.4%	
Working dog cross (Kelpie/Cattle dog/Koolie)	906	16.2%	10.1%	
Border Collie cross	312	5.6%	7.9%	
Bull Terrier cross	1327	23.8%	11.4%	
Bulldog cross	109	2.0%	8.3%	
Designer poodle cross	59	1.1%	2.9%	
Giant breed cross (Wolfhound/Mastiff/Dane)	527	9.4%	11.4%	
Large breed cross (German Shepherd/Ridge)	712	12.7%	9.5%	
Medium breed cross (Husky/Malamute/Vizsla)	306	5.5%	10.2%	
Purebred	153	2.7%	8.7%	
Small toy cross (Chihuahua/Maltese/Terrier)	587	10.5%	6.2%	
Pooled	5587			

^1.^ Estimated failure values at 90 days after adoption from multivariable models reported in Table 3 and Table 4. ^2.^ 2.6% if hazards for all age categories ≥3 months were reduced to those for dogs aged 2 to <3 months and the hazards for dogs aged 1 to <2 months were unchanged from those reflected in the hazard ratio reported in Table 5. ^3.^ 3.5% if hazards for all breeds were reduced to those for small toy cross (Chihuahua/Maltese/Terrier) other than for designer poodle cross dogs where hazards were unchanged from those reflected in the hazard ratios reported in Table 6; 6.8% if hazards for all breeds were reduced to those for designer poodle cross dogs.

## Data Availability

No new data were created. All data were from existing RSPCA Queensland records and Australian Bureau of Statistics (ABS; https://www.abs.gov.au/statistics accessed on 19 September 2022). Considerable work was done in manipulating the data to be in a form suitable for analyses and the dataset may be useful to others. Data are available on request from the corresponding author.

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
