# Peer review of "Adoption Can Be a Risky Business: Risk Factors Predictive of Dogs Adopted from RSPCA Queensland Being Returned"

_animals, 2022, doi:10.3390/ani12192568_

Round 1
Reviewer 1 Report
This is a very nicely done and well written manuscript. My comments are minor and most are just a bit of clarification. My first overall comment is a bit philosophical in that in some countries we’ve accepted that returns are a part of adoption—and that a readmission can be a learning opportunity for the shelter. I do think that even the very fast ones could be viewed in this light—especially for dogs who haven’t been in foster the adopter can provide valuable information that can improve the dog’s chance of adoption and retention next time. I think that this perspective could be mentioned early on (around the section which mentions a forever home) as well as in the discussion where the cost benefit of the interventions to prevent returns are presented. This idea can help shelters consider what they can learn from their readmissions when an intervention just isn’t practical. My other concerned comment is about the grey highlighting: I really like it but can’t figure out the pattern. Sorry! See my specific comments below.
Line 88-9: forever home is a bit outdated and implies that readmission is always a bad thing. It may not be when situations change and the household can’t or shouldn’t keep the dog. Please edit.
Please edit in several highly visible places that refunds for adoption fees only happen within the first 14 days. That is likely an important driver for that time period (although not for the day 1 and 2 returns).
Line 114: unique id number: was this consistently applied to a dog each time that dog came back? How did the shelter know if was one of their dogs (was this via microchip)?
Line 151: was this including neuter status at intake or only at adoption?
Line 205: body weight at intake or adoption? And what happens for puppies?
Line 260: starting “The distribution table…” is a sentence fragment.
Section 3.3: I would love to see the median and min/max time to readmission for the stray, seized, euthanized, and rehomed groups.
I think that 2 significant figures for p-values makes the point without extra reading and interpretation. And for HR as well unless the 3rd digit is really important? I’d also prefer the % to be rounded as it is easier to read and take them in that way.
Table 1: adoptions is missing a letter. Any sense that some of these findings could be due to lack of power given the rarity of some of the results, e.g., # of prior adoptions with only 2 dogs readmitted for the 2-4 category?
Line 303: please explain in the text what council admissions are.
Table 2: there are a few rows like days available for adoptions 10 to <15 where the p-value was < 0.05 and the CI excluded 1 that are not shaded. Any reason for that? Please edit as needed. Same for Table 3. See my overall comment at the top…I can’t see that all significant predictive categories are highlighted and only those categories.
Lines 320-1: days in foster care 1-<7 was not significantly different from not fostered. Is there a reason that wasn’t used as the cut off? Does it make any difference? I could see the sweet spot for foster care being a week minimum rather than none or any. Please address in the manuscript.
Line 350-51 the initial parenthesis is missing.
Table 3: please add something in the title about including weight and not breed for orientation.
Table 4: I’m assuming that the general pattern for the variables that aren’t shown is similar to Table 3?
Line 397: there were 2 different models in tables 3 & 4, one for color and one for breed, no? How is this one model?
Line 414: and these PA estimates are also for the 90 day period, not the shorter, higher risk time…please add that caveat here for clarity.
Line 490: “from” is missing the f.
Line 560: or that people aren’t going to be returning these dogs from these shelters…it could be a real result in this geography.
Paragraph starting line 562: The authors have an interesting spin on the impact of variables which can’t be changed, per se. I think that it might be wise to include a brief comment about what these variables might be proxy for as it seems like colour was addressed and yet, why would this inherently result in returns? There must be some other (perhaps latent) variable? Or do the authors believe it is due to misclassification?
Limitations section: if these results don’t generalize, why publish? I believe this should be published but this section is a bit too blunt and unhelpful. I might adjust it to indicate where the authors believe the results might be generalizable, with care.
Reviewer 2 Report
Please see the attached document for my comments.
